# The Properties of Micro Carbon Fiber Composite Modified High-Viscosity Asphalts and Mixtures

**DOI:** 10.3390/polym14132718

**Published:** 2022-07-02

**Authors:** Quanman Zhao, Shuo Jing, Xiaojin Lu, Yao Liu, Peng Wang, Min Sun, Li Wang

**Affiliations:** School of Transportation Engineering, Shandong Jianzhu University, Jinan 250101, China; jingshuo99@163.com (S.J.); luxiaojin0329@163.com (X.L.); liuyao199704@163.com (Y.L.); wangpeng@sdjzu.edu.cn (P.W.); sunmin@sdjzu.edu.cn (M.S.); wangli@sdjzu.edu.cn (L.W.)

**Keywords:** road engineering, ultra-thin overlay, micro carbon fiber, styrene butadiene rubber, road performance

## Abstract

In order to produce a high-viscosity asphalt and mixtures that can be used for ultra-thin overlays, high contents of Styrene–butadiene–styrene (SBS, 5%, 6%, 7%), styrene butadiene rubber (SBR, 1%, 2%, 3%) and micro carbon fiber (MCF, 0.8%) were used to modify conventional asphalt to prepare high-viscosity modified asphalt suitable for this purpose. The performance of the modified asphalts was evaluated by conventional index, kinematic viscosity, dynamic shear rheological test (DSR), multiple stress creep recovery test (MSCR), and bending beam rheometer test (BBR). The road performance of the modified asphalt mixtures was evaluated by high-temperature rutting, low-temperature bending, freeze-thaw splitting, fatigue, speckle, anti-skid, and water seepage tests. The results show that increasing the content of SBS can improve the high-temperature deformation resistance, low-temperature failure strain, kinematic viscosity, and viscosity toughness of modified asphalt, and the optimum content of SBS was 6%. SBR can improve the high-temperature performance, kinematic viscosity, and water damage resistance of modified asphalt, and the optimum dosage was 2%. Compared with 5% SBS-modified asphalt mixture, the dynamic stability, low-temperature failure strain, and freeze-thaw splitting strength ratio of 6% SBS + 0.8% MCF composite-modified asphalt mixture were increased by 48.7%, 24.7%, and 5.2% respectively. Compared with the 5% SBS-modified asphalt, the same characteristics of the 2% SBR + 5% SBS + 0.8% MCF composite-modified asphalt increased by 127.1%, 13.5%, and 5.5%, respectively. Compared with 5% SBS-modified asphalt, the fatigue performance of 6% SBS + 0.8% MCF-modified asphalt was improved by 32.2%. The kinematic viscosity of 6% SBS + 0.8% MCF and 5% SBS + 0.8% MCF + 2% SBR modified asphalt met the performance requirements of high-viscosity asphalt and had excellent road performance. It can be applied to ultra-thin overlays to optimize its adhesion with the original pavement.

## 1. Introduction

As of early 2022, total highway mileage had reached 5.2807 million kilometers in China, of which maintenance mileage accounted for 99.4%, so the task of road maintenance is arduous [1]. To reduce road damage, increase the service life of road surfaces, reduce road surface overhaul, and save resources, it is of great eco-environmental value and engineering application significance to build ultra-thin overlays to maintain pavement in response to the national “double carbon” policy. In the 1990s, ultra-thin overlays began to be used in preventive maintenance technology for asphalt pavement, and the overlay thickness was reduced to 15–20 mm [2]. The reduction in thickness resulted in problems such as quick aging, poor durability, and poor water stability of the pavement [3,4,5]. In order to prolong the life cycle of ultra-thin overlays, it is recommended to use high-viscosity modified asphalt to improve its road performance. Therefore, the improvement of modified asphalt viscosity and mixture performance has been studied by many scholars [6,7,8].

Studies have found that the composite modification of asphalt by adding different modifiers can effectively improve its performance and meet the road performance requirements of ultra-thin overlays [9,10]. The road performance of ultra-thin overlays is highly dependent on the adhesion of the asphalt. For instance, the rheological properties of Rubber/SBS composite-modified asphalt and Crumb rubber/SBS composite-modified asphalt were studied, and it was found that, compared with conventional asphalt, the modified asphalt showed a significant improvement in viscoelasticity and viscosity, and road performance [11,12]. In addition, asphalt mixture properties can be improved by nanomaterials. It is reported that the binding energy of the asphalt binder was enhanced by nano-montmorillonite (MMT), and the adhesion to the aggregate was increased, so the self-healing ability of microcracks was improved [13]. Besides, some new technologies have been developed to enable composite materials to have many functional properties such as flame retardance, anti-corrosivity, electrical conductivity, etc. [14,15,16,17,18].

Carbon fiber is an excellent modifier that can improve the road property of asphalt mixtures and significantly improve asphalt toughness. In recent years, many scholars have studied carbon nanotubes and carbon fibers as modifiers to prepare modified asphalt for ultra-thin overlays [19,20,21,22]. The anti-aging properties, self-healing properties, bending properties, and fracture toughness of asphalt mixtures can be improved by different fiber modifiers [23,24,25,26]. In addition, the preparation of composite-modified asphalt by different modifiers and fibers helps to improve the road performance of ultra-thin overlays [27,28]. Although carbon fiber has excellent performance as a modifier, its cost is too high and the economic benefits of applying it to ultra-thin overlays are not good. As a new material with high strength and high modulus, MCF has the advantages of less defects, large specific surface area, large aspect ratio, low density, high modulus, high strength, and strong electrical and thermal conductivity.

Therefore, aiming to develop a modified asphalt with high toughness, high strength, and high viscosity for ultra-thin overlays, SBS/MCF and SBR/SBS/MCF composite-modified asphalt were proposed, and asphalt mixtures were prepared. The performance of the modified asphalts is evaluated by conventional index, kinematic viscosity, rotational viscosity, dynamic shear rheometer, multiple stress creep recovery, and low-temperature trabecular tests. The road performance of the composite-modified asphalt mixtures is evaluated through high-temperature rutting, low-temperature bending, freeze-thaw splitting, fatigue, speckle, anti-skid, and water seepage tests, then compared with SBS-modified asphalt. The effect of improving the performance of ultra-thin overlays is studied, the optimum amount of modifier is determined, and a composite-modified asphalt and mixture suitable for ultra-thin overlays are proposed.

## 2. Materials and Methods

### 2.1. Materials

#### 2.1.1. Raw Materials

Qinhuangdao 70^#^ base asphalt was selected as the raw material for modified asphalt. According to the “Technical Specification for Highway Asphalt Pavement Construction” (JTG E20-2011), the penetration, softening point, and ductility of asphalt were measured [29]. Table 1 shows various performance indicators. Styrene butadiene styrene (SBS) produced by PetroChina Dushanzi Petrochemical Company (Dushanzi, Xinjiang, China) was selected, and its technical indicators were provided by suppliers, as shown in Table 2. The SBR of model 1502 produced by Shandong Xianyuan Chemical Technology Co., Ltd. (Zibo, Shandong, China) was selected. Various indicators of MCF were provided by suppliers, as shown in Table 3.

#### 2.1.2. Preparation of Modified Asphalt

First, the matrix asphalt was preheated to 160 °C. Second, the extraction oil (mass fraction of 2%) was mixed into the matrix asphalt and heated to 180 °C in a thermal insulation jacket. It was then cut with a high speed shear and kept at a speed at 4000 rpm, SBS (mass fraction of 5%, 6%, 7%), SBR (mass fraction of 1%, 2%, 3%), and MCF (quality score of 0.8%) were slowly added, and the shearing time lasted for 1 h. It was found by previous research that the optimal dosage of MCF was 0.8%, so in this experiment, the dosage of MCF was 0.8%. Finally, the stabilizer was added at 180 °C and stirred for 3 h (rotation speed 650 rpm) to gain the composite-modified asphalt. 5% SBS-modified bitumen was used as a comparison sample.

#### 2.1.3. Aggregate

The coarse aggregate was basalt-crushed stone with a particle size of 5–10 mm. The fine aggregate was basalt rock debris with a particle size of 0–3 mm. The ore powder was limestone ore powder. Table 4 shows the relevant technical parameters.

#### 2.1.4. Asphalt

According to the test results of asphalt performance, 5% SBS, 0.8% MCF + 5% SBS + +2% SBR, 0.8% MCF + 5% SBS, 0.8% MCF + 6% SBS, and 0.8% MCF + 7% SBS-modified asphalt were used as binders to prepare the asphalt mixture.

#### 2.1.5. Modified Asphalt Mixture Gradation

According to the engineering characteristics of ultra-thin overlays, SMA-10 was selected for the gradation type. Drawing on previous laboratory mix design experience and technical specifications [30], the SMA-10 asphalt mixture gradation was designed, as shown in Table 5.

After the primary selection and gradation, the optimum oil-to-stone ratio was determined to be 7.3% by the Marshall test. The volume index of the asphalt mixture was calculated, and the results are shown in Table 6, all of which meet the requirements of the specification [29].

#### 2.1.6. Tests for Modified Asphalt

(1)Penetration, Softening Point, Ductility, viscosity, and toughness tests
According to the test procedure [29], FLUKO brand high-speed shearing machine, FLUKO brand electric mixer, SYD-0620A asphalt kinematic viscosity tester, asphalt penetration tester, asphalt softening point tester, and asphalt ductility tester were used to test modified asphalt: 60 °C Kinematic viscosity, viscosity, and toughness.

(2)Brookfield viscosity
Rotational Brookfield Viscosity Test according to JTG E-2011 “Highway Engineering Asphalt and Asphalt Mixture Test Regulations” T 0625 [29]. The measurement temperatures were 100 °C, 120 °C, 135 °C, 155 °C, and 175 °C, the rotor model was 27^#^. The change of the viscosity and temperature relationship of the asphalt material was used to analyze its high-temperature flow characteristics.

(3)Dynamic shear rheological test (DSR)
A temperature sweep test was carried out on the modified asphalt, 52–88 °C as test temperature range, the temperature increase rate was 2 °C/min, the fixed strain was 0.5%, and the frequency was 1 HZ. The variation laws of rutting factor with temperature were compared and analyzed.

(4)Multiple stress creep recovery test (MSCR)
Under the conditions of 0.1 kpa and 3.2 kpa stress levels and a test temperature of 64 °C, 10 cycles were tested at each stress level. One cycle was 10 s, which comprised the creep loading stage of 1 s and the unloading recovery stage of 9 s.

(5)Bending beam rheometer test (BBR)
Preparation and test operation of BBR were carried out according to specification JTG E20-2011 [29] and American AASHTO M320-10. The test temperatures were −12 °C, −18 °C and −24 °C.

The specific research process is shown in Figure 1.

#### 2.1.7. Tests for Asphalt Mixture

(1)Rutting test
According to the standard test methods JTG E20-2011, The rutting experiments of SBS, SBS/MCF and SBS/SBR composite-modified asphalt mixture were carried out at 60 °C [29]. The experimental equipment is shown in Figure 2.

(2)Low temperature bending test
The low temperature bending failure test used a trabecular with a length of 250 mm, a width of 30 mm and a height of 35 mm, and its span was 200 mm. At a temperature of −10 °C, and at a speed of 50 mm/min, loading at a single point in the middle of the span was used to calculate the failure of the trabecular and maximum bending strain. The experimental equipment is shown in Figure 3.

(3)Freeze-thaw splitting test
According to the regulations JTG E20-2011 [29], cylindrical specimens were made, and the water stability of the asphalt mixture was evaluated by freeze-thaw splitting test. The experimental equipment is shown in Figure 4.

(4)Four-point bending fatigue test and digital image correlation test
Before the four-point bending fatigue test, the static load test was conducted. The static load test adopted the displacement control mode and loaded at a speed of 0.01 mm/s until the specimen broke and failed, and bending tensile strength was obtained.

(5)The bending fatigue test
The bending fatigue test adopted the load control mode, and the loading waveform was an uninterrupted asymmetric equal-amplitude sine wave. The cyclic eigenvalue (that was, the ratio of Fmin to Fmax) was taken as R = 0.1, where the maximum cyclic loading was the product of the stress level S and the ultimate bearing capacity, and the minimum loading was the product of the maximum and the cyclic eigenvalue R. The loading frequency was 10 Hz, and the loading was stopped when the specimen was fractured. The specimen was tested with three stress levels: 0.3, 0.4, and 0.5. Before the fatigue test started, first, the specimen was preloaded, and loaded linearly to the load value corresponding to the median value of the sine wave within 30 s to ensure that the test indenter and the specimen were pressed tightly. Holding the load constant for 20 s, the DIC collected the first image at the same time.

The DIC equipment adopted the Xi’an Xintuo three-dimensional full-field strain measurement system. The equipment consists of an image acquisition and analysis system, two high-precision TAWOV cameras, a plane calibration board, a control box, a high-performance workstation and a blue light source, and the hardware and software are realized through the control box. The signal link of the TAWOV camera was about 80 mm in length, the lens focal length was 25 mm, the resolution was 2448 × 2048, the pixel count of the camera was 5 million, and the pixel size was 3.45 µm × 3.45 µm. The fixed angle between the two cameras was 25 degrees. The equipment performed a series of strain monitorings and analyses of the asphalt mixture.

The test loading waveform, test process and DIC equipment are shown in Figure 5.

(6)Anti-skid and water seepage test
The friction coefficients of SBS, SBS/MCF, and SBS/SBR/MCF-modified asphalt were tested according to the test method for pavement friction coefficient (T0964-2008) by pendulum instrument in “Highway Subgrade Pavement Field Test Regulations” (JTG E60-2008), and their skid resistance in wet conditions were evaluated. The structural depths of SBS, SBS/MCF, and SBS/SBR/MCF-modified asphalt were tested according to the manual sand laying method in the “Site Test Regulations for Highway Subgrade Pavement” (JTG E60-2008).

## 3. Results and Discussion

### 3.1. Three Indexes and Viscosity and Toughness

The test results of SBS, SBS/MCF, and SBS/SBR/MCF composite-modified asphalts are displayed in Table 7.

The data results in Table 7 show that with the increase of the SBS content, the penetration gradually decreased, and the softening point, ductility, 60 °C Kinematic viscosity, viscosity, and toughness gradually increased. It shows that SBS was evenly dispersed in the modified asphalt, interacts with the asphalt, and forms a stable spatial network structure, which improved the high and low temperature performance and temperature sensing performance of the asphalt. The softening point and 60 °C kinematic viscosity of SBS/MCF-modified asphalt showed an upward trend when the content of SBR increased. When the SBR content exceeds 2%, the increase of penetration will adversely affect the performance of modified asphalt. The viscosity and toughness both increased first and then decreased. As the SBR absorbs the oil content in the asphalt and swells, the colloidal structure of the asphalt was improved, and the viscosity was increased. From the analysis of the experimental results, it can be found that increased the content of SBS can improved the high and low temperature performance, viscosity, and toughness of asphalt. Added SBR can improved the high-temperature performance of SBS/MCF-modified asphalt, and the comprehensive performance of composite-modified asphalt was the best when the SBR content was 2%.

Analysis of the above experimental data, 5% SBS (comparative sample), SBS (5%, 6%, 7%) + 0.8% MCF and 5% SBS + 0.2% SBR + 0.8% MCF were selected as the best asphalt mixtures and their rheological properties were studied.

### 3.2. Rheological Properties of the Composite-Modified Asphalts

#### 3.2.1. Brookfield Viscosity

Viscosity is crucial for the construction and workability of modified asphalt. The Brookfield viscosity of modified asphalt in the temperature range of 100–175 °C was tested. The test results are shown in Figure 6. By analyzing the data results in Figure 6 and comparing the viscosity-temperature curves, it can be concluded that the viscosity of modified asphalt decreased when the temperature increases, and the viscosity difference between modified asphalts with different dosages becomes smaller. It is more sensitive to the amount of SBS when the temperature decreased. The kinematic viscosity shows a trend of increasing gradually when the content of SBS increased, which shows that SBS can improve the viscosity of modified asphalt. When the content of SBR was 2%, the viscosity of the composite-modified asphalt was the highest. Compared with SBS-modified asphalt and SBS/MCF-modified asphalt, the viscosity of SBS/SBR/MCF-modified asphalt at 135 °C increased by 182% and 167%, respectively. It shows that the kinematic viscosity and high-temperature deformation resistance of SBS/MCF-modified asphalt were significantly improved by SBR. Although the increase in the content of SBS makes the viscosity of the composite-modified asphalt gradually increase, it will also increase the construction temperature of the mixture, which will generate a large amount of flue gas and cause pollution. Therefore, the content of SBS in the composite-modified asphalt should not be too high [31,32].

#### 3.2.2. High-Temperature Rheological Property

The test results of the dynamic shear rheological are shown in Figure 7.

The analysis results in Figure 7 show that the rutting factor G*/sinδ of the modified asphalt decreased gradually with the increased of temperature, and the higher the temperature, the smaller the difference in the rutting factor between different modified asphalts. At the same temperature, the G*/sinδ of the modified asphalt was gradually increased with the increased of the SBS content. This trend indicated that the high-temperature deformation resistance of the modified asphalt was improved by the increase of the SBS production, and it increased first and then decreased with the increase of the content of SBR. When the content of SBR was 2%, G*/sinδ reached the maximum value, indicating that adding an appropriate amount of SBR can improve the viscosity of the composite-modified asphalt, thereby improving its high-temperature rutting resistance.

#### 3.2.3. Elastic Resilience

The high-temperature rheological properties of modified asphalt were studied by MSCR test to test the elastic recovery rate R and the non-recoverable creep compliance Jnr [33]. Under the elastic capacity, the non-recoverable creep compliance was used to evaluate the rutting resistance of the asphalt. The larger the R and the smaller the Jnr, the better the high-temperature stability of the asphalt mixture. The MSCR test results are shown in Figure 8 and Table 8.

It can be seen from Figure 8 that the cumulative strain changed with time in the same way under the two stress levels, the cumulative strain increased gradually with time, and decreased with the increase of SBS content. This shows that increasing the content of SBS can improve the high-temperature deformation resistance of modified asphalt. When the SBR content increased, the accumulated strain shows a trend of first decreasing and then increasing. The accumulated strain reaches the lowest value when the SBR content is 2%. The analysis shows that the high-temperature rheological properties of asphalt can be significantly improved by SBR. The composite-modified asphalt with 5% SBS + 2% SBR + 0.8% MCF had the strongest high-temperature deformation resistance.

Analyzing the data in Table 8, it can be seen that *J_nr_* gradually decreased, and R increases gradually after the content of SBS increased under the two stress levels, indicating that the high-temperature stability and elastic properties of asphalt are improved by increasing the content of SBS. With the increase of SBR content, the Jnr value first decreased and then increased, and the R value first increased and then decreased. When the SBR content was 2%, the Jnr value was the smallest and the R value was the largest. This shows that adding SBR can significantly improve the elasticity and rutting resistance of asphalt at high-temperature.

#### 3.2.4. Low-Temperature Creep Properties

The low temperature performance indexes of the mixture with stiffness modulus S and creep rate m were obtained by BBR test. The test results are shown in Figure 9.

Figure 9 shows that the stiffness modulus S of the modified asphalt increases gradually with decreasing temperature, and the creep rate m gradually decreased, indicating that the modified asphalt gradually became hard and brittle as the temperature decreased, and stress relaxation capacity and low temperature performance are reduced. At −18 °C, the stiffness modulus S of 6% SBS + 0.8% MCF was the smallest, and the creep rate m was the largest, indicating that when the SBS content was 6%, the low-temperature performance was the best. The stiffness modulus S of 5% SBS + 2% SBR + 0.8% MCF composite-modified asphalt compared with 5% SBS + 0.8% MCF composite-modified asphalt at −12 °C, −18 °C, −24 °C, respectively, reduced by 34.8%, 25.5%, 2.0%, and creep rate m increased by 6.4%, 0.6%, 1.9%, respectively, indicating that the low temperature performance of modified asphalt can be improved by adding SBR.

## 4. Asphalt Mixture Road Performance

### 4.1. High-Temperature, Low-Temperature, and Water Stability Test

The dynamic stability, failure strain, and freeze-thaw splitting strength ratio were obtained through rutting, freeze-thaw splitting tests, and low-temperature bending failure. The test results are shown in Table 9.

From Table 9, the DS of 5%, 6%, 7% SBS + 0.8% MCF mixtures increased by 11.5%, 20.4%, and 48.7%, respectively, compared with the mixtures of 5% SBS, indicating that the high-temperature performance of asphalt mixture can be improved by increasing the content of SBS. Compared with 5% SBS + 0.0% MCF and 7% SBS + 0.8% MCF, the DS of the mixture after adding 2% SBR increased by 127.1% and 37.0%, respectively, indicating that the high-temperature stability and rutting resistance of the asphalt mixture were significantly improved after adding SBR.

By comparing the failure strains in Table 9, the low temperature failure strain of 6% SBS + 0.8% MCF, 5% SBS + 2% SBR+ 0.8% MCF was increased by 24.7% and 13.5%, respectively, compared with 5% SBS. It shows that the low temperature cracking resistance of asphalt mixture can be improved by adding SBR and increasing the content of SBS, and the 6% SBS + 0.8% MCF asphalt mixture had the best low-temperature crack resistance.

Based on the above test results, 5% SBS (comparative sample), 6% SBS + 0.8% MCF, 5% SBS + 2% SBR + 0.8% MCF were further selected for four-point bending fatigue, anti-skid and water seepage tests.

### 4.2. Fatigue Property

The fatigue performance of the modified asphalt mixtures was evaluated by four-point bending fatigue test in stress control mode [34,35,36]. The bending stress-deflection curve and fatigue test results of the specimens under static load are shown in Figure 10.

A series of strain monitoring and analyses of the asphalt mixtures was carried out using DIC equipment [37,38,39]. The 6% SBS + 0.8% MCF asphalt mixture was loaded 20,000 times under the condition of 0.3 stress ratio, and the strain cloud diagram is shown in Figure 11. Fatigue bending strain analysis was performed on the mixtures at stress ratios of 0.3, 0.4 and 0.5, as shown in Figure 12.

Analysis of Figure 10a shows that under the condition of static loading, the deflection increased gradually, and the bending stress first increased and then decreased. The maximum failure stress value of 6% SBS + 0.8% MCF was the largest, indicating that it had the best resistance to damage.

Analysis of Figure 10b shows that the fatigue life greatly decreased with the increase of the stress ratio, and a higher stress level seriously affected the fatigue life of the asphalt mixture. Under the same stress level, the fatigue performance of asphalt mixture with 6% SBS + 0.8% MCF content was the best and 5% SBS + 2% SBR + 0.8% MCF was the worst. At 0.3, 0.4, and 0.5 stress levels, the fatigue life of 6% SBS + 0.8% MCF was increased by 17.4%, 13.0%, and 32.2%, respectively, compared with 5% SBS-modified asphalt, indicating that increasing the content of SBS and adding MCF can improve asphalt mixing. The addition of SBR will adversely affect the fatigue properties of asphalt mixtures.

Analysis of Figure 11 shows that micro-cracks first generated at the initial stage of loading, and when the deformation accumulated to a certain extent, macro-fine cracks began to appear at the relatively weak position at the bottom of the specimen. With the increase of loading times, the cracks developed gradually. At the same time, the specimen can withstand the load cycle without breaking after the crack occurs because of the toughening effect of the fiber and the bonding effect of the asphalt. Due to the connection of the fibers, the cracks did not develop vertically upward along the original path, but tortuously cracked, indicating that the addition of micro-carbon fibers can significantly improve the fatigue performance of asphalt mixtures.

The analysis in Figure 12 shows that, at three different stress levels, the strain level of the modified asphalt mixtures gradually increased as the number of loading cycles increased. The maximum principal strain value of 6% SBS + 0.8% MCF was higher than for the other two materials, indicating that the 6% SBS + 0.8% MCF material had the best deformation resistance.

### 4.3. Anti-Slip and Anti-Water Seepage Properties

The test results of the friction coefficient, structural depth, and water permeability coefficient of the mixture are shown in Table 10.

It can be seen from Table 10 that the friction coefficient BPN of the three modified asphalt mixtures all meet the requirements of the specification [29] (≥80), indicating that the three modified asphalts have good anti-skid performance. The friction coefficient increases after the addition of MCF and SBR, which indicates that the surface of the basalt aggregate can be evenly coated by the composite-modified asphalt, and the adhesion of the composite-modified asphalt was improved. 5% SBS + 2% SBR + 0.8% MCF has the largest friction coefficient and the best skid resistance. The structural depths of the three modified asphalt mixtures were all greater than the specification requirements [29] (≥0.5 mm), indicating that the three modified asphalts have good anti-skid performance. 0.8% MCF + 6% SBS and 0.8% MCF + 5% SBS + 2% SBR compared with 5% SBS-modified asphalt structure depth increased by 36.1% and 44.4%, respectively, indicating that the increase of modified asphalt viscosity can improve the mixture structure depth. The water permeability coefficients all meet the requirements [29] (≤200 mL/min), indicating that the water sealing effect of the mixtures was good, and the composite-modified asphalt mixtures meet the waterproof performance requirements of the surface layer.

## 5. Conclusions

The main conclusions of this paper are summarized as follows:(1)Based on the evaluation of DSR, BBR, and MSCR tests, the high and low temperature properties and the viscosity of the prepared modified asphalt were significantly improved compared with the SBS-modified asphalt. The optimal dosage combination was 6% SBS + 0.8% MCF, 5% SBS + 2% SBR + 0.8% MCF.(2)According to the results of rutting, low-temperature trabecular bending failure, freeze-thaw splitting, four-point bending fatigue, and speckle tests, the prepared modified asphalt can improve the high-temperature stability, low-temperature crack resistance, water stability, and fatigue performance of an asphalt mixture, and the performances of 6% SBS + 0.8% MCF and 5% SBS + 2% SBR + 0.8% MCF-modified asphalt mixture were the best.(3)The anti-skid and water seepage tests show that the modified asphalt prepared in this study provided excellent and durable skid resistance for the ultra-thin overlays, which greatly increases driving safety.

Finally, this study shows that 0.8% MCF + 6% SBS and 0.8% MCF + 5% SBS + 2% SBR composite-modified asphalt and mixtures have excellent high and low temperature performance, viscosity toughness, and fatigue resistance, which can meet the requirements of ultra-thin coating requirements, and can be used for ultra-thin overlays.

## Figures and Tables

**Figure 1 polymers-14-02718-f001:**
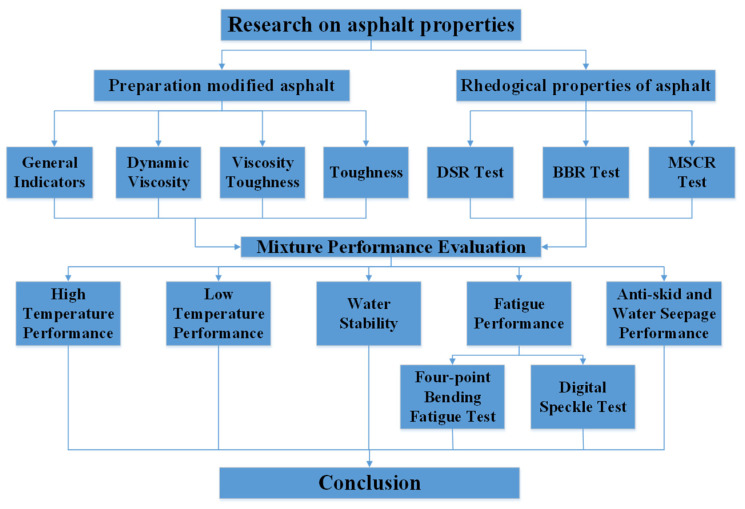
Design of the experiments.

**Figure 2 polymers-14-02718-f002:**
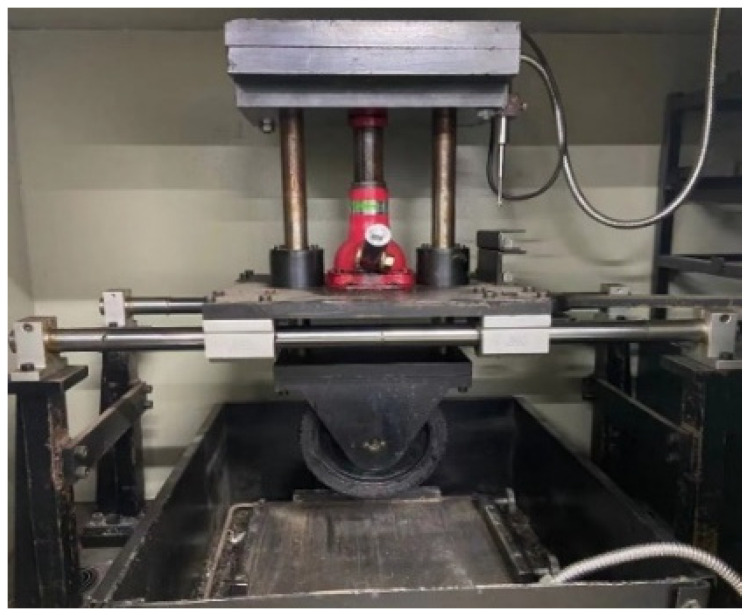
Rutmeter.

**Figure 3 polymers-14-02718-f003:**
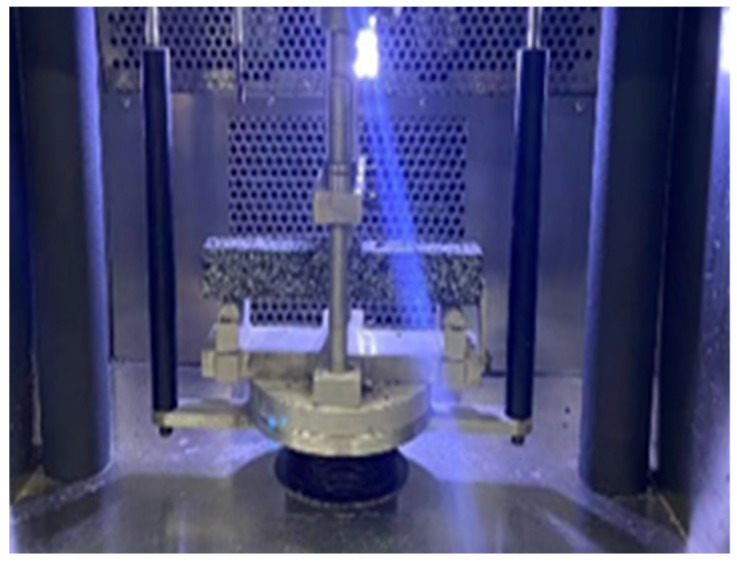
Multifunctional testing machine.

**Figure 4 polymers-14-02718-f004:**
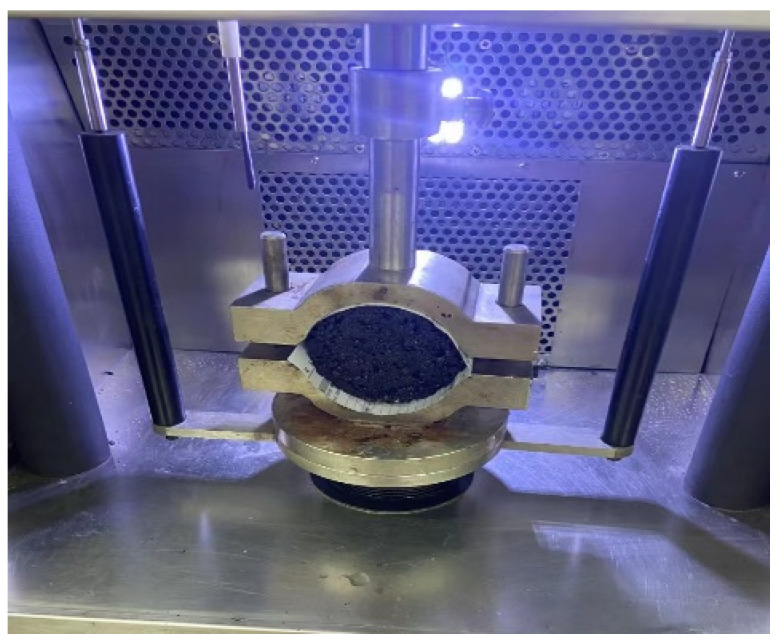
Multifunctional testing machine.

**Figure 5 polymers-14-02718-f005:**
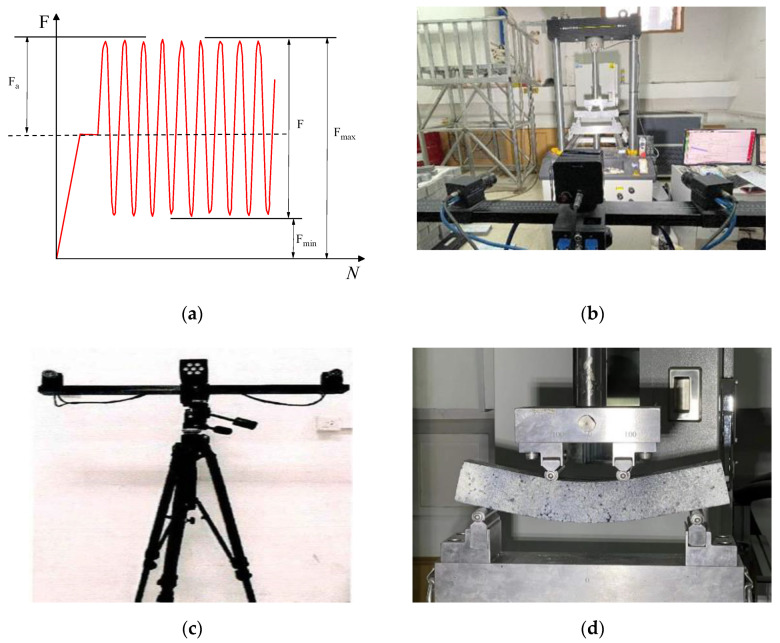
Bending fatigue test. (**a**) Load waveform; (**b**) Experimental procedure; (**c**) DIC equipment; (**d**) Bending state of the specimen.

**Figure 6 polymers-14-02718-f006:**
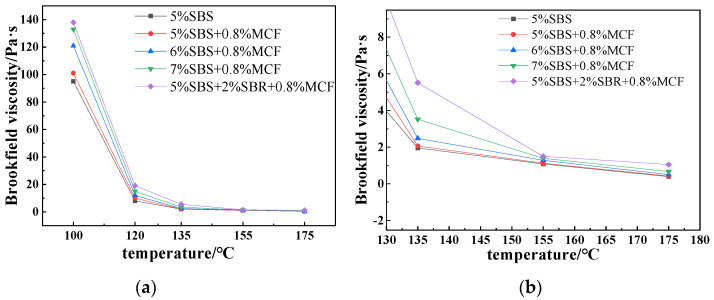
Rotational viscosity test results for the modified asphalts. (**a**) 100–175 °C temperature range; (**b**) 130–175 °C temperature range.

**Figure 7 polymers-14-02718-f007:**
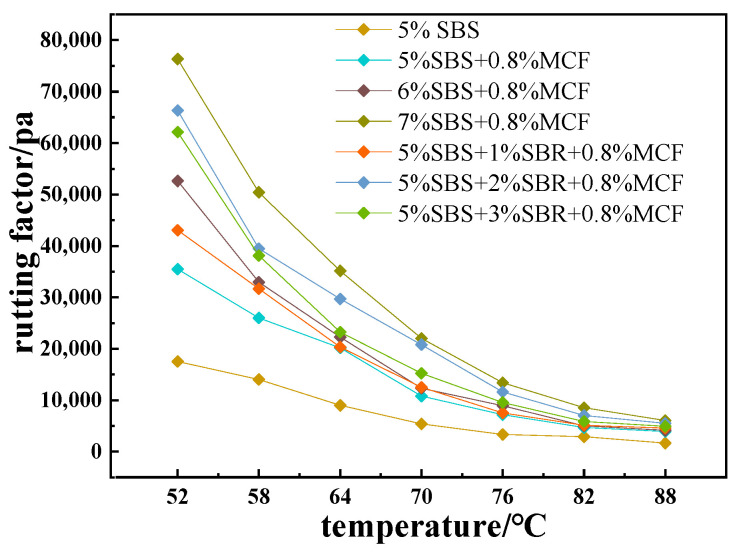
Rutting factor of composite-modified asphalt.

**Figure 8 polymers-14-02718-f008:**
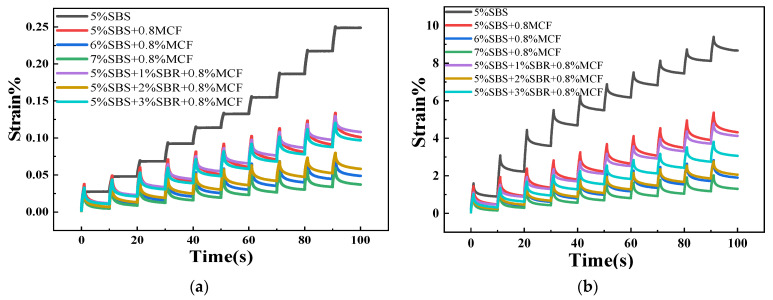
Strain change of asphalt material. (**a**) 0.1 kPa; (**b**) 3.2 kPa.

**Figure 9 polymers-14-02718-f009:**
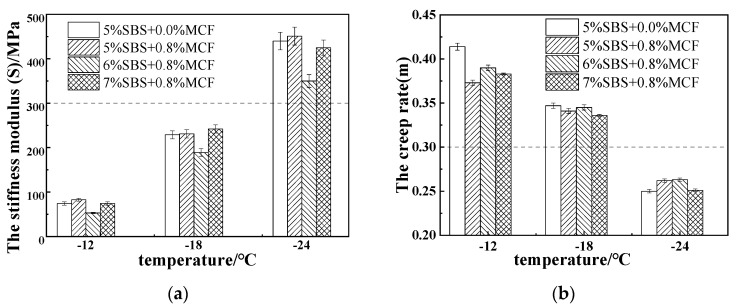
Low-temperature trabecular test results of modified asphalt at different temperatures: (**a**) Test results for S value; (**b**) Test results for m value.

**Figure 10 polymers-14-02718-f010:**
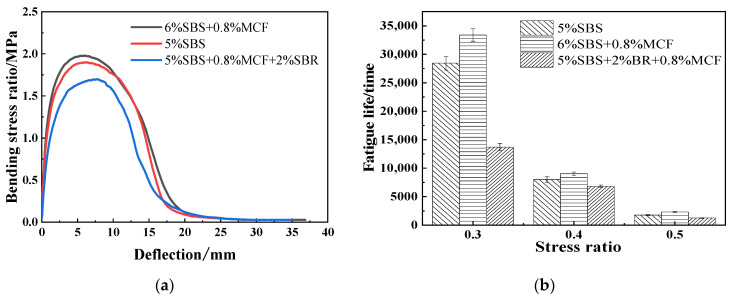
Bending stress-deflection curves and fatigue test results: (**a**) Bending stress- deflection curve; (**b**) Results of fatigue tests for different stress levels.

**Figure 11 polymers-14-02718-f011:**
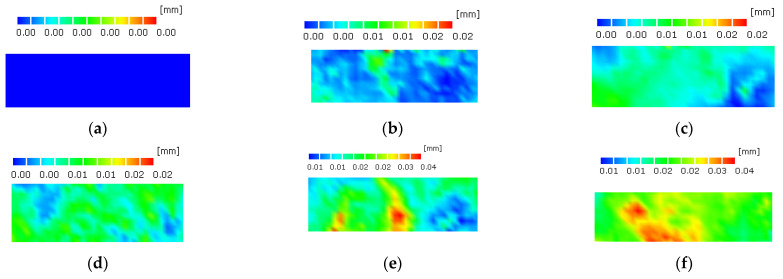
Strain cloud diagram of 6% SBS + 0.8% MCF asphalt mixture under different loading times in fatigue test. (**a**) loaded 0 times; (**b**) loaded 4000 times; (**c**) loaded 8000 times; (**d**) loaded 12,000 times; (**e**) loaded 16,000 times; (**f**) loaded 20,000 times.

**Figure 12 polymers-14-02718-f012:**
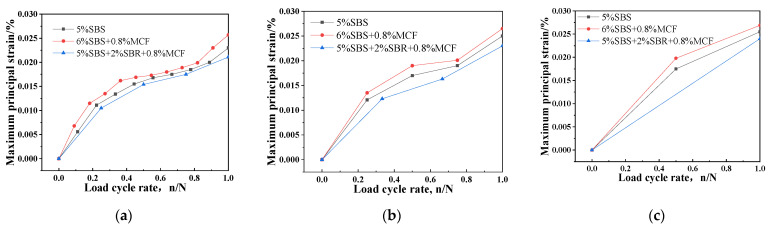
Relationship between maximum principal strain and loading cycle rate: (**a**) Relationship curves under 0.3 stress ratio; (**b**) Relationship curves under 0.4 stress ratio; (**c**) Relationship curves under 0.5 stress ratio.

**Table 1 polymers-14-02718-t001:** Technical indexes of base asphalt.

Performance Indicators	Penetration (25 °C)/0.1 mm	Ductility (10 °C)/cm	Softening Point/°C	Flash Point/°C	Wax Content/%	Density (15 °C)/(g·cm^−3^)
Test results	63.8 ± 0.3	52 ± 2	47 ± 1	320 ± 5	1.5 ± 0.2	1.032 ± 0.003

**Table 2 polymers-14-02718-t002:** SBS material performance index.

Index	Reference
Block Ratio(S/B)	30/70
Tensile Strength/MPa	>18.0
Elongation/%	>600
Permanent Deformation/%	<40
Shore Hardness(A)	≥70
Ash/%	0.1
Volatile Composition/%	≤0.5

**Table 3 polymers-14-02718-t003:** Technical indexes of micro carbon fiber.

Index	Specification Type (mm)	Density (g/cm^3^)	Tensile Strength (MPa)	Modulus of Elasticity (GPa)
Data results	0.017	1.8	4900	230

**Table 4 polymers-14-02718-t004:** Summary of aggregate density of each grade.

Material Type	Relative Apparent Density	Relative Gross Volume Density	Water Content (%)	Asphalt Absorption Coefficient	Relative Effective Density (g/cm^3^)
Basalt 5–10 mm	2.9307	2.8194	1.3511	0.60	2.8859
Basalt 0–3 mm	2.8689	2.8689	-	0.93	2.8689
Mineral Powder	2.6335	2.6335	-	-	2.6335

**Table 5 polymers-14-02718-t005:** SMA-10 Composite gradation.

Sieve Size (mm)	Passing Percentage of Each Sieve Hole (%)								
	13.2	9.5	4.75	2.36	1.18	0.6	0.3	0.15	0.075
Composite Gradation	100.0	98.7	43.9	25.2	21.8	18.6	16.3	14.0	12.1
SMA Specification Gradation Upper Limit	100	100	60	32	26	22	18	16	13
SMA Specification Gradation Lower Limit	100	90	28	20	14	12	10	9	8
SMA Specification Gradation Median	100	95	44	26	20	17	14	12.5	10.5

**Table 6 polymers-14-02718-t006:** SMA-10 Marshall test results.

Marshall Indexes	Gross Volume Density (g/cm^3^)	Maximum Theoretical Density (g/cm^3^)	Volume of Air Void/%	Voids in Mineral Aggregate (%)	Voids Filled with Asphalt (%)	Stability (kN)	Flow Value (mm)
Test Results	2.452	2.555	4.0	20.3	80.1	16.5	4.9
Specification Requirements	-	-	3~4	≥17.0	75~85	≥6.0	-

**Table 7 polymers-14-02718-t007:** Test results of asphalt technical indicators.

Modified Asphalt Type	Penetration (0.1 mm)	Softening Point (°C)	Ductility (mm)	60 °C Kinematic Viscosity (kPa·s)	Viscosity Toughness (N·m)	Toughness (N·m)
5% SBS	53.77 ± 1.03	74.70 ± 0.20	29.50 ± 0.75	16.00 ± 1.00	24.60 ± 0.65	16.0 ± 0.20
0.8% MCF + 5% SBS	52.40 ± 0.37	77.70 ± 0.50	33.50 ± 2.50	22.00 ± 1.00	30.00 ± 1.30	22.0 ± 0.00
0.8% MCF + 6% SBS	50.77 ± 0.23	79.10 ± 0.30	34.20 ± 1.40	25.00 ± 1.00	32.00 ± 1.50	23.7 ± 0.35
0.8% MCF + 7% SBS	46.30 ± 0.35	82.55 ± 0.20	34.80 ± 0.85	31.25 ± 0.25	34.60 ± 2.00	24.6 ± 0.45
0.8% MCF + 5% SBS + 1% SBR	48.17 ± 0.57	76.15 ± 0.20	29.00 ± 2.50	15.30 ± 0.60	30.00 ± 0.20	20.4 ± 0.00
0.8% MCF + 5% SBS + 2% SBR	46.17 ± 0.27	80.15 ± 0.40	32.50 ± 1.75	42.00 ± 1.00	31.30 ± 0.40	21.2 ± 0.60
0.8% MCF + 5% SBS + 3% SBR	47.40 ± 0.57	80.70 ± 0.00	27.85 ± 0.50	59.85 ± 1.15	31.23 ± 1.53	18.9 ± 0.25

**Table 8 polymers-14-02718-t008:** MSCR test results of the modified asphalts.

Asphalt Type	5% SBS + 0.0% MCF	5% SBS + 0.8% MCF	6% SBS + 0.8%MCF	7% SBS + 0.8% MCF	5% SBS + 1% SBR + 0.8% MCF	5% SBS + 2% SBR + 0.8% MCF	5% SBS + 3% SBR + 0.8% MCF
*J_nr_*_0.1_/kPa^−1^	0.25 ± 0.01	0.10 ± 0.01	0.05 ± 0.00	0.04 ± 0.00	0.10 ± 0.00	0.06 ± 0.00	0.09 ± 0.01
*J_nr_*_3.2_/kPa^−1^	0.27 ± 0.01	0.13 ± 0.00	0.06 ± 0.00	0.04 ± 0.00	0.10 ± 0.01	0.07 ± 0.00	0.09 ± 0.00
*R*_0.1_/%	46.91 ± 0.12	75.10 ± 0.10	85.17 ± 0.13	85.68 ± 0.10	65.17 ± 0.08	77.92 ± 0.10	70.17 ± 0.09
*R*_3.2_/%	47.07 ± 0.08	70.38 ± 0.12	82.81 ± 0.12	84.54 ± 0.10	61.15 ± 0.11	96.36 ± 0.12	70.40 ± 0.10

**Table 9 polymers-14-02718-t009:** High-temperature, low-temperature, and water stability test results.

Mixture Type	DS/(Time·mm^−1^)	Failure Strain $\varepsilon_{B}/\mu \varepsilon$	*TSR* (%)
5% SBS	5045 ± 26	2867 ± 20	90.6 ± 0.3
0.8% MCF + 5% SBS	5625 ± 16	3226 ± 15	91.3 ± 0.2
0.8% MCF + 6% SBS	6774 ± 28	3578 ± 31	93.5 ± 0.1
0.8% MCF + 7% SBS	8362 ± 17	3447 ± 15	95.3 ± 0.1
0.8% MCF + 5% SBS + 2% SBR	11,455 ± 25	3256 ± 23	95.6 ± 0.2

**Table 10 polymers-14-02718-t010:** Results of anti-slip and water seepage tests.

Mixture Type	Friction Coefficient BPN	Depth (mm)	Water Permeability Coefficient (mL/min)
5% SBS	81 ± 2	0.72 ± 0.05	76 ± 2
6% SBS + 0.8% MCF	85 ± 2	0.98 ± 0.06	77 ± 1
5% SBS + 2% SBR + 0.8% MCF	87 ± 1	1.04 ± 0.05	80 ± 2

## Data Availability

This original copy does not include distributed figures and tables, thus all figures and tables in this original copy are unique.

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
