# Peer review of "The Properties of Micro Carbon Fiber Composite Modified High-Viscosity Asphalts and Mixtures"

_polymers, 2022, doi:10.3390/polym14132718_

Round 1

Reviewer 1 Report

The paper seeks to introduce an approach ‘’ The properties of MCF composite modified high-viscosity asphalts and mixtures. However, the authors should consider improving upon the quality to further highlight and emphasis. 

1.    Introduce one or two lines highlighting the problem you are trying to solve at the beginning of the abstract

2.    Put space between each value and its corresponding units. Consider spacing between the values and their percentage units.

3.    State clearly in one or two lines at the end of the abstract, the significance of the study.

4.    The introduction needs to be improved by relating to the mechanics of the studied materials and their mechanical characteristics. The references to be included are: 10.1007/s10853-022-06994-3, 10.1016/j.polymertesting.2017.09.009, 10.1016/j.compstruct.2021.114698, 10.1177/0731684417727143 and 10.1002/app.46770.

5.     In section 2.1.3, point 2, you stated that “the rotor model was 27#”. Is the # part of the model definition. If yes, explain, and if not, then correct it.

6.    The font size of the figure is too small. Increase it to 14 or more visible

7.    Learn when to apply hyphens to words. For instance, “viscosity-temperature” and “high-temperature”. What rule warrants a hyphen. Such anomalies run through the whole manuscript.

 Check the spellings well. The paragraph beneath the figure has the word figure written as “figyre 3” instead of figure 3.

Reviewer 2 Report

The paper entitled “The properties of MCF composite modified high-viscosity asphalts and mixtures” by Zhao et al., describes the asphalt properties due to adding SBS, SBR, and MCF with various concentrations. The manuscript contains promising data but needs major revision before consideration for publication in the Polymers journal. 

1.     The title should not contain abbreviations, please revise the title.

2. Abstract, abbreviations such as SBS and SBR should be mentioned the complete first time, please revised throughout the manuscript.

3. Line 21, 23, 25, and 26, please add space between the symbol or number and units, please check and revise throughout the manuscript

4. Please add a clear hypothesis at the end of the introduction followed by how to confirm or achieve this hypothesis.

5. The authors divided the material and methods into two-part “2. Modified Asphalt Materials and Methodology” and “4. Modified Asphalt Mixture Materials and Methodology”, why? Authors can combine all material and methods under one title “Material and Methods” followed by various subtitles.

6. Therefore, the title “2. Modified Asphalt Materials and Methodology” should be changed to “2. Material and methods”.

7. Line 109, the authors said, “Table 1 shows various performance indicators”, this data is according to reference or what, also, data in Tables 2 and 3, please clarify.

8. Line 128, “(1) Three major indexes, viscosity, and toughness tests”, these two indices, please revised.

9. Line 139, 144, and 148, please add the complete name of these abbreviations.

10. Data in the current investigation is not statistically analyzed, please check and revise.

11.  Line 160 – 170, please explain the obtained results, such as decreasing the penetration by increasing the SBS content, whereas, increasing other factors.

12.  Figures 5 and 10b, please add standard error according to the statistical analysis of data.

13.  The manuscript needs a deep discussion of the obtained data.

14.  The manuscript contains various typo-error, please revise the language carefully.

Reviewer 3 Report

The paper presents an investigation on the properties of MCF composite modified high-viscosity asphalts and mixtures

The following recommendations are proposed:

·         Please, rewrite the abstract. The reviewer believes that this can be organized to provide a wider audience with a better understanding of the work performed.

·         The introduction can be restructured.

·         I recommend introducing the research approach in a different section.

·         Some parts of the figures are blurry.

·         Overall, English needs to be double-checked for typos.

·         Conclusions can be reorganized for clarity purposes.

Main concern:

What is the novelty of this paper?

Round 2

Reviewer 2 Report

The authors revised the manuscript as per the suggested points. This form of the manuscript can be accepted for publication.

Reviewer 3 Report

·        Comments addressed